# Non-Insulin Antidiabetic Agents and Lung Cancer Risk in Drug-Naive Patients with Type 2 Diabetes Mellitus: A Nationwide Retrospective Cohort Study

**DOI:** 10.3390/cancers16132377

**Published:** 2024-06-28

**Authors:** Tomasz Tabernacki, Lindsey Wang, David C. Kaelber, Rong Xu, Nathan A. Berger

**Affiliations:** 1Center for Artificial Intelligence in Drug Discovery, Case Western Reserve University School of Medicine, Cleveland, OH 44106, USA; 2Center for Science, Health, and Society, Case Western Reserve University School of Medicine, Cleveland, OH 44106, USA; 3Center for Clinical Informatics Research and Education, The MetroHealth System, Cleveland, OH 44109, USA; 4Case Comprehensive Cancer Center, Case Western Reserve University School of Medicine, Cleveland, OH 44106, USA

**Keywords:** lung cancer, type 2 diabetes mellitus, insulin therapy, non-insulin antidiabetic agents, glucagon-like peptide-1 receptor agonists, cancer risk reduction, retrospective cohort study

## Abstract

**Simple Summary:**

Lung cancer is the leading cause of cancer-related deaths in the United States, and people with Type 2 Diabetes often use insulin therapy, which has been associated with increased lung cancer risk. This study explores whether non-insulin diabetes medications can lower this risk compared to insulin. By analyzing medical records from over a million patients, the researchers aim to identify which medications are associated with a reduced risk of lung cancer compared with insulin. The results could help providers select treatments that not only manage blood sugar but also minimize lung cancer risk, leading to better patient outcomes and informing future treatment guidelines.

**Abstract:**

Lung cancer (LC) is the second most common cancer and the leading cause of cancer deaths in the U.S. Insulin therapy, a key treatment for managing Type 2 Diabetes Mellitus (T2DM), is associated with increased LC risk. The impact of non-insulin antidiabetic drugs, particularly GLP-1 receptor agonists (GLP-1RAs), on LC risk is not well understood. This study evaluated LC risk in T2DM patients, comparing seven non-insulin antidiabetic agents to insulin. Using the TriNetX Analytics platform, we analyzed the de-identified electronic health records of 1,040,341 T2DM patients treated between 2005 and 2019, excluding those with prior antidiabetic use or LC diagnoses. We calculated hazard ratios and confidence intervals for LC risk and used propensity score matching to control for confounding factors. All non-insulin antidiabetic drugs, except alpha-glucosidase inhibitors, were associated with significantly reduced LC risk compared to insulin, with GLP-1RAs showing the greatest reduction (HR: 0.49, 95% CI: 0.41, 0.59). GLP-1RAs were consistently associated with lowered LC risk across all histological types, races, genders, and smoking statuses. These findings suggest that non-insulin antidiabetic drugs, particularly GLP-1RAs, may be preferable for managing T2DM while reducing LC risk.

## 1. Introduction

Lung cancer (LC) is the second most common malignancy in the U.S. in both men and women and the leading cause of cancer mortality in both sexes [1]. Strategies for controlling modifiable risk factors are critically needed. Insulin therapy remains a cornerstone in Type 2 Diabetes Mellitus (T2DM) management, prescribed to millions worldwide. However, recent studies have suggested a potential connection between insulin therapy and an increased incidence of LC [2,3,4]. This association may stem from mechanisms such as insulin resistance, hyperinsulinemia, insulin promotion of LC growth, and the promotion of an inflammatory microenvironment [5]. In preclinical models, insulin has been shown to foster LC growth through pathways related to these mechanisms, and clinical outcomes in LC patients receiving insulin therapy have been observed to be worse compared to those using other antidiabetic interventions [5,6].

Non-insulin antidiabetics have demonstrated a differential risk of LC in retrospective studies [7,8,9], though the evidence remains limited and sometimes controversial [10]. Metformin, the most widely prescribed antidiabetic drug, has been widely explored for its potential anticancer properties [5,11], while SGLT2 inhibitors are being explored for their potential in LC prevention [12]. Other antidiabetics, such as sulfonylureas, alpha-glucosidase inhibitors (AGIs), thiazolidinediones (TZDs), and dipeptidyl peptidase-4 (DPP-4) inhibitors, have been associated with varying levels of cancer risk in LC studies [13]. Among these, glucagon-like peptide 1 receptor agonists (GLP-1RAs) stand out for their wide-ranging benefits, including glycemic control, weight reduction, and immune modulation [14,15,16]. Recent findings indicate that GLP-1RA use is linked to a decreased colorectal cancer risk [17].

Despite these insights, the differential effects of antidiabetic treatments on LC risk in patients with T2DM have yet to be systematically examined and compared with insulins within a single clinical dataset utilizing unified methods. The absence of research comparing the impact of different antidiabetic medications on LC risk prevents the development of clear guidelines for optimal treatment of T2DM with comorbidities such as tobacco use, which may place them at increased risk for the development of LC. To address this gap, we conducted a nationwide, retrospective cohort study among drug-naive T2DM patients. Our research compares LC risk among users of GLP-1RAs and six other non-GLP-1RA antidiabetics with insulin users.

## 2. Methods

We used the TriNetX platform to access aggregated, de-identified electronic health records of 105.9 million patients, including 7.7 million with T2DM, from 61 healthcare organizations across 50 states, covering diverse geographical regions, age, ethnicity, income and insurance groups, and clinical settings [18]. TriNetX provides aggregated data from inpatient and outpatient clinical encounters, including demographics, diagnoses, procedures, medications, laboratory values, and vital statuses, mapping the data to a consistent clinical data model with a consistent semantic meaning so that the data can be queried consistently regardless of the underlying data source(s). TriNetX’s built-in analytical functions (for example, incidence, prevalence, outcomes analysis, survival analysis, and propensity score matching) allow for patient-level analyses while only reporting population-level data to the researcher. The MetroHealth System institutional review board determined that using data from TriNetX is not human-subject research and, therefore, is exempt from approval. The TriNetX platform has recently been shown to be useful for multiple population-based cohort studies, including cancer [17,19,20,21].

The study population comprised 1,040,341 patients with T2DM who had medical encounters for T2DM and were subsequently prescribed antidiabetic medications from 2005 to 2019, with no prior antidiabetic medication use (drug naive) and no prior LC diagnosis. The cohort was widely geographically distributed, with 32% in the Northeast, 18% in the Midwest, 42% in the South, and 7% in the West, with <1% unknown. GLP-1RAs, metformin, alpha-glucosidase inhibitors, DPP-4 inhibitors, SGLT2 inhibitors, sulfonylureas, and thiazolidinediones were all compared against insulin users. The study period of 2005 to 2019 (except for the starting year of 2013 for SGLT2 inhibitors and 2006 for DPP-4 inhibitors) was chosen based on the year drugs were first approved. The study population was divided into exposure cohorts and comparison cohorts for each comparison.

To mitigate confounding, cohorts were propensity score matched (1:1, using nearest neighbor greedy matching) for demographics, BMI, A1c, pre-existing medical conditions, family and personal history of cancers and lung nodules, lifestyle factors (e.g., smoking and alcohol use), environmental exposures, and procedures such as LC screening, which were determined via ICD-10 codes (Table 1). The outcome was the first diagnosis of LC that occurred within 15 years starting from the index event, also determined via ICD-10 diagnostic codes. The follow-up period for each patient began with the index event, which was the first prescription of insulins or the comparison medications, and ended with the occurrence of the outcome, death, the end of the patient’s medical record, or the end of the 15-year period. With censoring applied, hazard ratios (HRs) and 95% confidence intervals (CI) were calculated using Cox proportional hazards models. Secondary analyses were also conducted for the GLP-1RA versus insulin comparison, stratifying LC risk by histological subtype: squamous cell carcinoma, adenocarcinoma of the lung, large cell carcinoma, and small cell LC. Further subgroup analyses for the GLP-1RA vs. insulin comparison were performed, stratifying by race, ethnicity, sex assigned at birth, and smoking status determined by the presence or absence of relevant ICD-10 codes before the index event.

Data were collected and analyzed on 1 February 2024, within the TriNetX Analytics Platform using built-in functions (R, version 4.0.2 [R Project for Statistical Computing]), with statistical significance set at a 2-sided *p* < 0.05 [18]. This study followed the Strengthening the Reporting of Observational Studies in Epidemiology (STROBE) reporting guidelines [22].

## 3. Results

In the study cohort of 1,040,341 drug-naive individuals with T2DM without prior LC diagnosis, significant differences were observed between the exposure and comparison cohorts (non-insulin antidiabetic drugs versus insulin) across several parameters. These included differences in average glycated hemoglobin (A1c) levels, body mass index (BMI), demographic profiles, prevalence of pre-existing medical conditions, exposure to adverse factors, and personal and family history of LC, in addition to variations in alcohol and tobacco use rates. Notably, the cohort receiving glucagon-like peptide 1 receptor agonists (GLP-1RAs) in comparison to those on insulin therapy was characterized by a younger demographic, a higher average BMI, a greater proportion of females and Caucasians, and lower rates of tobacco and alcohol usage. Baseline characteristics for selected groups before and after matching are detailed in Table 1. Following the application of propensity score matching, a balance was achieved between each treatment and comparison group for all measured covariates, as evidenced by a standardized mean difference of less than 0.1. Complete baseline characteristics for all cohorts before and after matching are available in Appendix A.

During a 15-year follow-up in 1,040,341 drug-naive patients with T2DM, we observed a decreased risk for LC among all non-insulin antidiabetic drugs when compared with insulin except for alpha-glucosidase inhibitors. GLP-1RAs were associated with the largest reduction in LC risk compared to insulin (HR, 0.489, 95% CI, 0.408, 0.586), SGLT2 inhibitors (HR 0.598, 95% CI: 0.474, 0.755), and TZD (HR: 0.606, 95% CI: 0.513, 0.715) (Figure 1).

In further sub-group analysis, GLP-1RA users were associated with a significantly lower risk of incident LC when compared to insulin users across all histological types, including squamous cell carcinoma (HR: 0.364, 95% CI: 0.168, 0.789), adenocarcinoma of the lung (HR: 0.509, 95% CI: 0.333, 0.778), large cell carcinoma (HR: 0.345, 95% CI: 0.124, 0.959), and small cell LC (SCLCL) (HR: 0.101, 95% CI: 0.013, 0.8) (Figure 2). Significant reductions in LC risk were observed among Black (HR: 0.382, 95% CI: 0.206, 0.71) and White (HR: 0.511, 95% CI: 0.41, 0.638) patients but not in individuals of other races or in Hispanic individuals (Figure 3). Similar results were observed between males and females, as well as smokers and non-smokers (Figure 3).

## 4. Discussion

This study aimed to explore the association between antidiabetic drug use and differential LC risk among T2DM patients. On a worldwide basis, it is the leading cause of cancer-related mortality, contributing to 1.8 million deaths annually [23]. While tobacco use control has significantly led to a decreased incidence of LC in the U.S. [1,24], further methods for mitigating LC risk are needed. Given the rising prevalence of T2DM globally [25], understanding the potential effects of antidiabetic medications on LC risk is of great clinical relevance.

Our results indicate that all non-insulin antidiabetic drugs, excluding AGIs, are associated with a decreased risk of LC compared to insulins in antidiabetic-naive T2DM patients. GLP-1RAs were associated with the most significant reduction in LC risk (HR: 0.489), though the 95% confidence interval overlapped with that of SGLT2 inhibitors and TZDs. AGIs did not meet the threshold for significance, which may be attributed to the small sample size of just 1,441 individuals. Subgroup analysis highlighted GLP-1RAs’ efficacy across different LC histologies and was associated with a significantly decreased risk in Black and White patients, with limitations in other racial and Hispanic groups likely due to insufficient sample sizes. These findings suggest a substantial association between non-insulin antidiabetic medications, especially GLP-1RAs, and decreased LC risk compared to insulin therapy. These results may guide clinicians in selecting appropriate medications for diabetes management while mitigating the risk of developing LC. Our findings highlight that GLP-1RAs are associated with a significant reduction in LC risk compared with insulins, aligning with existing research suggesting that GLP-1RAs may be associated with a reduced risk of colorectal cancer [17]. A study comparing GLP-1RAs to metformin but not insulin showed an adjusted odds ratio of 0.81 of LC among GLP-1RA users [9]. Similarly, analysis of the SEER database indicates that DPP-4 inhibitors, which prevent the inactivation of endogenous GLP-1, are linked to improved survival in colon and lung cancers [26]. Despite these associations, there remains a scarcity of preclinical evidence directly elucidating the mechanisms by which GLP-1RA therapy may influence LC risk.

Several studies have suggested that insulin therapy could be associated with elevated cancer risk due to its mitogenic effects, potentially fostering the growth of certain cancer types [13,27,28,29,30]. Potential confounders to these findings have been widely discussed, namely, that insulin is often prescribed to patients with advanced T2DM or those with comorbid conditions, complicating direct comparisons with other medications [31]. To date, two large clinical trials have been conducted, which did not find significant increases in overall cancer risk with glargine insulin [32,33]. However, these trials were powered to assess cardiovascular outcomes, precluding definitive conclusions about the risk of specific malignancies [5]. In addition to studies suggesting a correlation between insulin therapy and an elevated risk of LC, several studies suggest worsened survival outcomes [2,3,4,6]. Hyperinsulinemia, which may occur in the context of insulin therapy, has been associated with increased LC development [34]. Insulin can stimulate the proliferation, migration, and drug resistance of NSCLC cells via the PI3K/Akt pathway [35]. Additionally, insulin has been shown to enhance LC survival by inhibiting proapoptotic cytokines [36]. Our study’s findings that non-insulin antidiabetic drugs, except for AGIs, were associated with a significantly lower LC risk than insulin therapy align with previous research evidence supporting that insulin therapy may promote a higher risk of LC.

One potential mechanism through which GLP-1RAs reduce LC risk includes the enhancement of innate immunity, which plays a crucial role in preventing tumorigenesis [16]. Specifically, GLP-1RA therapy has been observed to boost the functionality of natural killer (NK) cells, vital components of anti-tumor immune surveillance [37]. Additionally, GLP-1RAs have been shown to significantly improve biomarkers of inflammation and oxidative stress more effectively than standard antidiabetic treatments [38]. Given the susceptibility of both NSCLC and SCLC to immune-modulating therapies [39], these immunological effects may be particularly relevant in reducing LC risk.

Another possible explanation to consider is the pronounced weight loss facilitated by GLP-1RAs. Many cancers, such as colon, pancreatic, and renal cancers, are associated with obesity [40]. However, the relationship between BMI and LC risk is debated, with several meta-analyses suggesting an inverse relationship, though these results are not universally accepted [41]. Our study accounted for BMI variations through propensity score matching, and the observed reduction in LC risk associated with GLP-1RAs persisted regardless of BMI. This indicates that the beneficial impact of GLP-1RAs on LC risk may extend beyond their weight-loss effects alone.

In our study, both SGLT2 inhibitors and TZDs were associated with significant reductions in LC risk compared with insulins, with hazard ratios overlapping the 95% confidence intervals of GLP-1RAs. SGLT2 inhibitors, highlighted in research for their expression in metastatic LC tissues, have been associated with improved LC patient survival, though their anticancer mechanisms are still being defined [42]. These may include effects on glucose transport, mitochondrial function, and key signaling pathways, along with enhancing immune surveillance through PD-L1 suppression and increased T cell cytotoxicity [12].

Regarding TZDs, there have been reports of a reduced risk of LC among users, although the evidence remains mixed [43]. While some studies suggest a protective effect of TZDs against LC, meta-analyses have yet to confirm a significant difference in LC risk with TZD use [44]. The clinical implications of these findings for the management of LC risk in patients with diabetes necessitate cautious interpretation and further research to validate these associations and understand the underlying mechanisms.

Our study identifies several limitations that must be acknowledged to contextualize our findings on the decreased LC risk associated with non-insulin antidiabetic use compared to insulin. Firstly, our study’s observational nature makes it inherently susceptible to biases typical of such research methodologies. In all EHR databases, there is the possibility of inaccurate documentation of outcome diagnoses. As data from TriNetX are population-level, we were not able to independently verify individual cancer diagnoses through patient-level data points. The validity and completeness of cancer incidence ascertainment in TriNetX EHRs are unknown. However, as our study focuses on relative risk comparisons between groups on the same platform, these limitations do not have a substantial impact on our interpretation. While propensity score matching was employed to align cohorts on a range of factors—including demographics, BMI, A1c levels, medical history, and lifestyle factors—the reliance on EHR accuracy introduces the potential for unmeasured or uncontrolled confounders. These confounders could skew the observed associations, suggesting a need for a cautious interpretation of the decreased LC risk findings. Of note is that detailed information regarding the extent, duration, and amount of tobacco use is unavailable on the TriNetX platform.

Additionally, the limitations of EMR/EHR database research limit some aspects of our analysis. Incomplete data on medication dosing and duration of use in the TriNetX database prevented a comprehensive assessment of the effects of these variables on LC risk. As the TriNetX database does not allow for controlling events that occur after the index event, we are unable to account for the possibility that some individuals may have switched to another medication after their initial prescription. Similarly, incomplete data from ICD-O coding regarding histological subtypes limited the statistical power of our conclusions in these areas. The study’s exclusion of patients with prior antidiabetic medication use narrows the generalizability of our findings across the broader T2DM patient population. Additionally, many patients may receive multiple combinations of antidiabetic medications, which may alter their risk profile further. Nonetheless, the observed associations between antidiabetic medication use, particularly GLP-1RA, and LC risk reduction necessitate further investigation to confirm these findings and better understand the underlying mechanisms.

## 5. Conclusions

In conclusion, our study finds that GLP-1RAs and other non-insulin antidiabetic medications are associated with a lower risk of LC in patients with T2DM compared with insulins, providing information to guide clinicians in the selection of appropriate therapies for individuals with T2DM while mitigating the risk of developing LC. Investigating the underlying mechanisms that contribute to the lower LC risk observed with GLP-1RAs and other non-insulin antidiabetics is essential for a deeper understanding of these associations. Expanding the evidence base will be vital for refining treatment decisions and developing comprehensive care strategies for this patient population. Future studies should aim to elucidate the biological pathways involved, assess the impact of combination therapies, and validate these findings across diverse populations to ensure the broad applicability and effectiveness of these treatment modalities in reducing LC risk among T2DM patients.

## Figures and Tables

**Figure 1 cancers-16-02377-f001:**
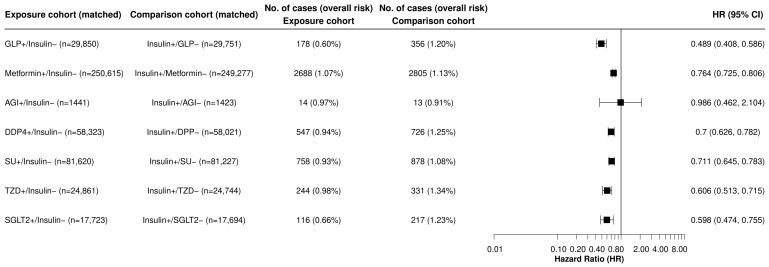
Analysis of Hazard Ratio comparing GLP-1RA and other anti-diabetic medications versus insulins Legend: Comparison of incident LC in the study population without previous LC and naive to antidiabetic medication (after the index event of the first prescription of non-insulin antidiabetic medication vs. insulin that occurred from 2005 to 2019). A plus sign (+) indicates that a patient was prescribed an antidiabetic medication, while a minus sign (−) indicates that they were not. AGI indicates alpha-glucosidase inhibitors; DPP-4, dipeptidyl-peptidase 4 inhibitors; SGLT2, sodium-glucose cotransporter-2 inhibitors; SU, sulfonylureas; TZD, thiazolidinediones; and GLP, glucagon-like peptide receptor-1 agonist.

**Figure 2 cancers-16-02377-f002:**
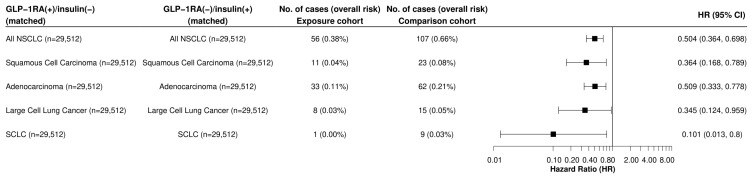
Comparison of GLP-1RA versus Insulins Across Lung Cancer Subtypes Legend: Comparison of incident LC stratified by lung cancer histological subtype within a 15-year time window after the index event of the first prescription of GLP-1RA medication vs. insulin that occurred from 2005 to 2019. The histological subtype was determined by ICD-O coding. A plus sign (+) indicates that a patient was prescribed an antidiabetic medication, while a minus sign (−) indicates that they were not. GLP-1RA: glucagon-like peptide receptor-1 agonist. NSCLC: non-small cell lung cancer; and SCLC: small cell lung cancer.

**Figure 3 cancers-16-02377-f003:**
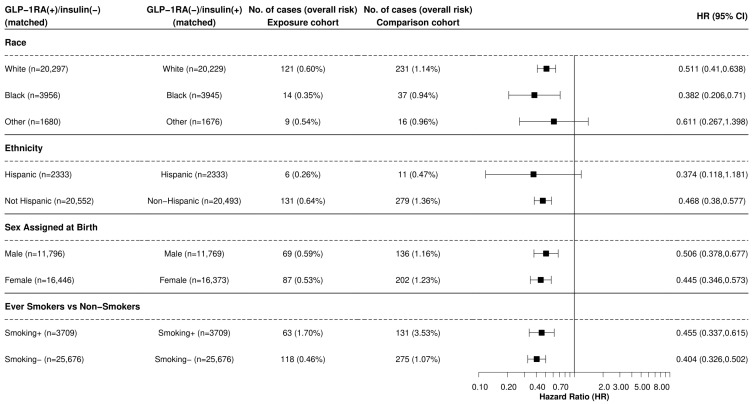
Comparison of GLP-1RA versus Insulins Stratified by Cohort Characteristics Legend: Comparison of incident LC stratified by demographic characteristics within a 15-year time window after the index event of the first prescription of GLP-1RA medication vs. insulin that occurred from 2005 to 2019. A plus sign (+) indicates that a patient was prescribed an antidiabetic medication, while a minus sign (−) indicates that they were not. GLP-1RA is a glucagon-like peptide receptor-1 agonist. “Other” groups include Asian, Native American, Alaskan Native, Native Hawaiian, and other Pacific Islanders. Ever-smoker status is defined by the presence of ICD-10 codes representing current or previous tobacco use.

**Table 1 cancers-16-02377-t001:** Demographic Characteristics of GLP-1RA and Metformin Exposure Cohorts versus Insulin Comparison Cohorts Before and After Propensity Score Matching.

	Before Propensity Score Matching, %		After Propensity Score Matching, %			Before Propensity Score Matching, %		After Propensity Score Matching, %		
	GLP-1RA (+)/Insulin (−) (*n* = 29,850)	Insulin (+)/GLP-1RA (−) (*n* = 628,808)	GLP-1RA (+)/Insulin (−) (*n* = 29,850)	Insulin (+)/GLP-1RA (−) (*n* = 29,751)	SMDc	Metformin (+)/Insulin (−) (*n* = 293,113)	Insulin (+)/Metformin (−) (*n* = 399,957)	Metformin (+)/Insulin (−) (*n* = 250,615)	Insulin (+)/Metformin (−) (*n* = 249,277)	SMD
Age at Index Event, mean (SD), y	**56.6 (11.6)**	**61.8 (15.7)**	56.5(11.6)	56.5(12.5)	0.0121	**59.7 (13.3)**	**62.5(16.4)**	60.5(13.3)	59.8(15.9)	0.0450
Sex										
Female	**54.9**	**47.8**	54.9	54.5	0.0077	**49.5**	**47.4**	48.9	48.8	0.0018
Male	**39.4**	**48.8**	39.4	39.8	0.0087	**46.6**	**48.8**	47.2	47.3	0.0016
Ethnicity										
Hispanic/LatinX	**7.8**	**9.2**	7.8	7.5	0.0098	**9.7**	**8.7**	9.5	9.6	0.0009
Not Hispanic/LatinX	**68.6**	**66.3**	68.6	69.0	0.0080	**67.1**	**65.5**	66.0	67.5	0.0321
Unknown	**23.6**	**24.5**	23.6	23.5	0.0026	**23.2**	**25.9**	24.5	23.0	0.0362
Race										
American Indian or Alaska Native	**0.4**	**0.4**	0.4	0.4	0.0018	**0.4**	**0.4**	0.4	0.4	0.0020
Asian	**2.5**	**4.2**	2.5	2.3	0.0134	**4.5**	**4.3**	4.3	3.9	0.0182
Black	**13.2**	**18.5**	13.2	13.6	0.0112	**16.7**	**17.7**	17.4	17.9	0.0148
Native Hawaiian or Other Pacific Islander	**0.5**	**1.3**	0.5	0.4	0.0120	**0.5**	**1.4**	0.6	0.6	0.0001
White	**68.1**	**61.4**	68.1	68.5	0.0079	62	62	61.9	62.0	0.0011
Unknown	**13.1**	**11.5**	13.1	12.7	0.0106	**12.6**	**11.9**	12.5	12.2	0.0069
Lifestyle Factors:										
Nicotine dependence	**7.5**	**13.2**	7.5	7.2	0.0118	**9.8**	**12.0**	10.4	10.1	0.0116
Personal history of nicotine dependence	**6.1**	**11.9**	6.1	5.5	0.0234	**5.5**	**12.9**	6.4	6.3	0.0045
Alcohol-related disorders	**1.3**	**3.8**	1.3	1.1	0.0139	**2.3**	**3.7**	2.6	2.4	0.0120
Family History and Screening										
Family history of malignant neoplasm of trachea, bronchus, and lung	**0.2**	**0.3**	0.3	0.2	0.0137	**0.2**	**0.4**	0.2	0.2	0.0043
Encounter for screening for malignant neoplasm of respiratory organs	**0.2**	**0.1**	0.2	0.1	0.0304	**0.1**	**0.1**	0.1	0.1	0.0059
Personal history of malignant neoplasm of bronchus and lung	**0.1**	**0.2**	0.1	0.1	0.0013	**0.1**	**0.4**	0.2	0.2	0.0018
Pre-Existing Health Conditions										
Asthma	**11.1**	**9.3**	11.1	10.8	0.0081	**8.8**	**8.5**	8.5	8.5	0.0008
Primary respiratory tuberculosis	0.0	<0.1	0.0	0.0		<0.1	<0.1	0.0	0.0	0.0012
Adverse Exposures:										
Pneumoconiosis due to other dust-containing silica	**<0.1**	**<0.1**	0.0	0.0	0.0000	**<0.1**	**<0.1**	0.0	0.0	0.0005
Contact with and (suspected) exposure to asbestos	**0.1**	**0.1**	0.1	0.1	0.0013	**0.1**	**0.1**	0.1	0.1	0.0030
Contact with and (suspected) exposure to air pollution	**<0.1**	**<0.1**	0.0	0.0	0.0258	**0.0**	**<0.1**	0.0	0.0	0.0089
Personal history of irradiation	**0.4**	**1.0**	0.4	0.3	0.0131	**0.4**	**1.1**	0.5	0.5	0.0041
Hemoglobin A1c, mean(SD),%	7.7 (1.8)	7.7 (2.2)	7.7 (1.8)	7.7 (2.0)		**7.2 (1.6)**	**7.5 (2.1)**	7.4 (1.7)	7.3 (1.9)	
>9%	**20.0**	**15.0**	20.0	19.7	0.0064	**10.0**	**11.9**	11.4	11.4	0.0002
<9%	**53.9**	**39.6**	53.9	53.6	0.0058	**54.4**	**35.4**	47.0	45.7	0.0277
No recorded A1c	**26.1**	**45.4**				**35.6**	**52.7**			
BMI, mean(SD), kg/m^2^	**36.3 (6.6)**	**31.5 (7.1)**	36.3 (6.6)	35.1 (6.5)		**33.5 (6.7)**	**30.9 (7.2)**	32.9 (6.8)	32.3 (7.0)	
0–18.4 kg/m^2^	**0.4**	**0.9**	0.4	0.3	0.0165	**0.4**	**1.0**	0.4	0.4	0.0017
18.5–24.9 kg/m^2^	**1.2**	**4.9**	1.2	1.1	0.0103	**2.6**	**5.2**	3.0	3.0	0.0017
25–29.9 kg/m^2^	**5.1**	**8.8**	5.1	4.9	0.0119	**7.4**	**8.2**	7.4	7.4	0.0025
At least 30 kg/m^2^	**18.8**	**14.4**	18.8	17.7	0.0278	**15.6**	**12.4**	14.1	13.4	0.0193
No recorded BMI	**74.5**	**71.0**				**74.0**	**73.2**			

The status of variables was based on the presence of related clinical codes anytime to 1 day before the index event. Bold indicates significant differences (*p* < 0.05) between exposure and comparison cohorts. A plus sign (+) indicates that a patient was prescribed the listed medication, while a minus sign (−) indicates that they were not. SMD is less than 0.1, a threshold indicating group balance. SMD, standardized mean difference.

## Data Availability

This study used population-level aggregate and de-identified data collected by the TriNetX Platform, which are available from TriNetX (https://trinetx.com accessed on 2 February 2024)); however, third-party restrictions apply to the availability of these data. The data were used under a license for this study, with restrictions that do not allow for the data to be redistributed or made publicly available. To gain access to the data, a request can be made to TriNetX (join@trinetx.com), but costs might be incurred, and a data-sharing agreement would be necessary. Data specific to this study, including diagnosis codes and group characteristics in aggregated format, are included in the paper as tables, figures, and Appendix A.

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
