# Peer review of "Non-Insulin Antidiabetic Agents and Lung Cancer Risk in Drug-Naive Patients with Type 2 Diabetes Mellitus: A Nationwide Retrospective Cohort Study"

_cancers, 2024, doi:10.3390/cancers16132377_

Round 1

Reviewer 1 Report

Comments and Suggestions for Authors

This is an interesting study with surprosing results.

Minor points:

1) Authors cite table 1 and 2 but I could only see Table 1 but not Table 2.

2) "We used the TriNetX platform to access deidentified electronic health records of 105.9  million patients, including 7.7 million with T2DM from 61 health care organizations across  50 states, covering diverse geographical regions, age, ethnicity, income and insurance  groups, and clinical settings. TriNetX built-in analytic functions allow for patient-level  analyses while only reporting population-level data."

Can authors rite something more about this plattform. What is the nature of the data? Outpatients or inpatients. They write about plattform but not about data themself.

What does "only reporting population-level data" mean?  Authors had separate patient records as else they could not conduct rgression analyses and macthed pairs.

3. Autjors correctly write in discussion tht OAD use was ASSOCIATED with a decreased LC risk, but in results and other parts they write "XX increased cancer risk".  Epidemiologically, such retropective studies can only show asociation and not a direct risk increase or decrease. Authors should check the whole manuscript to avoiding sentences using term "risk" without term "association".  THis should always be "associated with (for example) increased risk/decreased risk".

Reviewer 2 Report

Comments and Suggestions for Authors

Major comments:

1.     AGI has no effect to reduce LC risk compared to other groups (Figure1). How to explain the observation?

2.     The non-insulin treatments all can inhibit serum glucose levels, but mechanisms are not the same. Why all have the functions to reduce LC risk? In addition, GLP-1RA, which functions on increase insulin,  but, if insulin cause LC risk, why GLP-1RA is mor effect?

3.     Other types of non-insulin treatments can be shown across LC subtypes in Fig 2.

4.     Lung cancer cell line can be used to validate the findings, including insulin enhances LC, AGI has no effect compared to other treatments, GLP exhibits more benefit to reduce LC.

 Minor comments:

1.     Please check the sample number in Table 1 “GLP-1RA(+)/Insulin(-)” group in “Before Propensity Score Matching”.

Round 2

Reviewer 2 Report

Comments and Suggestions for Authors

Minor comments:

1.     Please check the sample number in Table 1 “GLP-1RA(+)/Insulin(-)” group in “Before Propensity Score Matching”.

2.     Why the figures become blurry in the revised manuscript?

Author Response

  1. Please check the sample number in Table 1 “GLP-1RA(+)/Insulin(-)” group in “Before Propensity Score Matching”.

Thank you for catching this again. I mistook your previous comment for a change in Supplementary Table 1. I have corrected the same typo in Table 1 now as well. 

  1. Why the figures become blurry in the revised manuscript?

We did not make any changes to the figures from the previous version. On our end they are displaying correctly. We will work with the publisher to ensure that the images are displayed appropriately.